# Cytostatic Action of Novel Histone Deacetylase Inhibitors in Androgen Receptor-Null Prostate Cancer Cells

**DOI:** 10.3390/ph14020103

**Published:** 2021-01-29

**Authors:** Zohaib Rana, Joel D. A. Tyndall, Muhammad Hanif, Christian G. Hartinger, Rhonda J. Rosengren

**Affiliations:** 1Department of Pharmacology and Toxicology, University of Otago, Dunedin 9016, New Zealand; ranzo073@student.otago.ac.nz; 2School of Pharmacy, University of Otago, Dunedin 9016, New Zealand; joel.tyndall@otago.ac.nz; 3School of Chemical Sciences, University of Auckland, Auckland 1142, New Zealand; m.hanif@auckland.ac.nz (M.H.); c.hartinger@auckland.ac.nz (C.G.H.)

**Keywords:** HDAC inhibitors, cancer chemotherapy, prostate cancer, pyridinecarbothioamide, metallodrugs, anticancer agents

## Abstract

Androgen receptor (AR)-null prostate tumors have been observed in 11–24% of patients. Histone deacetylases (HDACs) are overexpressed in prostate tumors. Therefore, HDAC inhibitors (Jazz90 and Jazz167) were examined in AR-null prostate cancer cell lines (PC3 and DU145). Both Jazz90 and Jazz167 inhibited the growth of PC3 and DU145 cells. Jazz90 and Jazz167 were more active in PC3 cells and DU145 cells in comparison to normal prostate cells (PNT1A) and showed a 2.45- and 1.30-fold selectivity and higher cytotoxicity toward DU145 cells, respectively. Jazz90 and Jazz167 reduced HDAC activity by ~60% at 50 nM in PC3 lysates. At 4 μM, Jazz90 and Jazz167 increased acetylation in PC3 cells by 6- to 8-fold. Flow cytometry studies on the cell phase distribution demonstrated that Jazz90 causes a G_0_/G_1_ arrest in AR-null cells, whereas Jazz167 leads to a G_0_/G_1_ arrest in DU145 cells. However, apoptosis only occurred at a maximum of 7% of the total cell population following compound treatments in PC3 and DU145 cells. There was a reduction in cyclin D1 and no significant changes in bcl-2 in DU145 and PC3 cells. Overall, the results showed that Jazz90 and Jazz167 function as cytostatic HDAC inhibitors in AR-null prostate cancer cells.

## 1. Introduction

Nearly 1.3 million new cases of prostate cancer were diagnosed in 2018, making it the second-most commonly diagnosed cancer and cause of fatalities in men [1]. Since 2005, a 66% increase in the incidence of prostate cancer has been observed due to an aging population [1,2]. Initially, prostate cancer is dependent on serum androgen levels, and therefore, androgen deprivation therapy is particularly effective [3]. However, after a median time of 18–24 months, the majority of patients develop castration-resistant prostate cancer (CRPC), where tumors grow in the absence of androgens, even though androgen receptors (ARs) are present [4,5]. Recently, studies have also reported that CRPC differentiates into a highly aggressive transdifferentiated AR-null/deficient/low heterogenous phenotype with or without a neuroendocrine phenotype [6,7,8]. Approximately 11–24% of patients develop this phenotype [6,9] and become resistant to enzalutamide, a Food and Drug Administration (FDA)-approved AR antagonist [6]. From a mechanistic viewpoint, abiraterone acetate and enzalutamide, which target the AR signaling pathways, are the only FDA-approved drugs for prostate cancers, but these drugs are not effective against AR-null prostate cancers [10]. Only one phase II trial using carboplatin and etoposide in AR-null tumors has been conducted. An objective response was only seen in 4 out of 56 patients according to the response evaluation criteria in solid tumors, and an overall survival of 9.6 months was reported [11]. Therefore, no FDA-approved treatment options are available for AR-null prostate cancer phenotypes.

Epigenetic modifications are a common feature of cancers, where gene expression is deregulated [12]. In these processes, acetylation and deacetylation of histones are critical in regulating gene expression [12]. Histone has an octamer structure flanked with tails. Following deacetylation by histone deacetylases (HDACs), the histone tails gain a positive charge and adhere to the negatively charged phosphate backbone of the DNA. In this state, gene transcription remains silent [13]. Histone acetyltransferases can add acetyl groups, resulting in the removal of positive charges. Consequently, the interaction between the phosphate backbone and histone tails are broken, leading to transcriptional activity [13].

HDACs are upregulated in a plethora of solid and hematological malignancies [13], which is linked to a poor clinical outcome [14]. Weichert and colleagues demonstrated that HDAC1, HDAC2, and HDAC3 are overexpressed in prostate cancers, whereas increased expression of HDAC1 and HDAC2 correlates with higher Gleason scores. The proliferative marker Ki67 showed a positive association with HDAC1, 2, and 3 [15]. Interestingly, studies report that HDAC1 has a repressive effect on the levels of androgen in prostate cancer cells, which in turn can be reversed by SENP1 expression [16]. Chiao and colleagues (2009) also showed that the AR-null cell lines PC3 and DU145 possess three to four times higher HDAC activity in comparison to the AR+ (LnCaP) cell line [17]. Therefore, HDAC inhibitors are likely to be more useful in an AR-null environment in comparison to an AR+ environment. The poor efficacy of HDAC inhibitors in clinical trials for CRPC are likely to be attributed to the lack of assessment of AR expression. HDAC inhibitors (romidepsin, Vorinostat or suberoylanilide hydroxamic acid (SAHA), panobinostat, and pracinostat) are well tolerated by patients and have been FDA approved for hematological malignancies, and an HDAC inhibitor (chidamide) has cleared phase III clinical trials in combination with exemestane for locally advanced or metastatically recurrent hormone receptor-positive breast cancer [18,19,20,21,22].

We have previously synthesized the novel HDAC inhibitors *N*1-hydroxy-*N*^8^-(4-(pyridine-2-carbothioamido)phenyl)octanediamide (Jazz90) and [chlorido(η^5^-pentamethylcyclopentadienyl)(*N*1-hydroxy-*N*^8^-(4-(pyridine-2-carbothioamido-κ^2^*N,S*)phenyl)octanediamide)rhodium(III)] chloride (Jazz167) (Figure 1), which are structural analogues of SAHA [23]. Substitution of the phenyl group of SAHA for a pyridinecarbothioamide moiety yielded Jazz90. It was then coordinated to Rh(pentamethylcyclopentadienyl)Cl to obtain Jazz167. Jazz90 and Jazz167 consist of an additional aromatic ring in comparison to SAHA, which could potentially improve the lipophilicity of these compounds, ultimately leading to an improved volume of distribution [24]. Alongside HDAC1, 6, and 8 inhibition, Jazz90 and Jazz167 possessed cytotoxic activity against colorectal, lung, and cervical cancer cells. Minimal hemolytic activity (10%) was recorded at high concentrations of 200 μM. In comparison, cisplatin, an FDA-approved metal-based drug for bladder cancer, exhibited 100% hemolytic activity at 200 μM [23]. Similarly, organometallic pyridinecarbothioamide compounds featuring osmium and ruthenium centers have also exhibited cytotoxicity against CH1 (ovarian), SW480 (colorectal), and A549 (lung) cancer cells [25].

Due to the role of HDAC overexpression in prostate cancer, the effects of Jazz90 and Jazz167 were investigated in CRPC. They were examined in AR+ (LnCaP) and AR− (DU145 and PC3) prostate cancer cells and non-cancerous (PNT1A) cells. This study explored the cancer cell to normal cell selectivity, HDAC inhibitory potential, and acetylation induction of the novel compounds. We also investigated the cytostatic potential versus the cytotoxic effect of Jazz90, Jazz167, and SAHA in prostate cancer cells to further understand their mechanism of action and clinical potential.

## 2. Results

### 2.1. Cytotoxic Potency

To determine the cytotoxic potential of the novel HDAC inhibitors Jazz90 and Jazz167, a range of prostate cancer cells were treated for 72 h. Jazz90, Jazz167, and SAHA had EC_50_ values of 2.09, 2.39, and 1.11 µM in PC3 cells, respectively (Appendix A and Table 1). In DU145 cells, EC_50_ values of 2.50, 6.44, and 0.69 µM were obtained for Jazz90, Jazz167, and SAHA, respectively (Table 1). No selectivity of HDAC inhibitors was seen for AR+ (LnCaP) cells, as similar EC_50_ values of 3.04, 3.36, and 0.61 µM were observed for Jazz90, Jazz167 and SAHA, respectively. Although SAHA expressed lower EC_50_ values in cancer cell lines in comparison to Jazz90 and Jazz167, it had a lower selectivity index (SI) of 1.65 compared to the selectivity indices of 2.93 and 3.50 for Jazz90 and Jazz167 in PC3 cells (Appendix A and Table 1). SAHA had the highest SI (3.07), followed by Jazz167 (2.53) and Jazz90 (2.02) in LnCaP cells. SAHA was most active against DU145 cells (2.71), followed by Jazz90 (2.45) and Jazz167 (1.30). The EC_50_ values in NIH 3T3, a preneoplastic cell line, were comparable to PNT1A cells, where Jazz167 exhibited a selectivity of 0.50, followed by Jazz90 (0.64) and SAHA (1.33) (Table 1).

### 2.2. HDAC Inhibition and Cellular Effects on Acetylation of Histone-3 Variant

To determine if Jazz90 and Jazz167 could inhibit a range of HDACs, HDAC inhibition assays were conducted in nuclear lysates from both HeLa and PC3 cells. Both Jazz90 and Jazz167 exhibited similar HDAC inhibition to SAHA (Figure 2). Jazz90 led to HDAC inhibition of 34 and 45% at 0.1 and 0.2 μM, whereas Jazz167 resulted in 38 and 53% inhibition at 0.1 and 0.2 μM, respectively, in HeLa cells. SAHA led to HDAC inhibition of 44 and 59% at 0.1 and 0.2 μM, respectively, in HeLa cells (Figure 2a). The positive control (trichostatin A (TSA)) led to 80% HDAC inhibition at a concentration of 0.1 μM (Figure 2a). In the PC3 lysate, Jazz90 resulted in inhibition of 58 and 69% compared to 59 and 64% inhibition for Jazz167 and 57 and 63% for SAHA (Figure 2b) at 0.05 and 0.1 μM, for all compounds, respectively.

To validate the results from HDAC inhibition assays, molecular docking was conducted. The crystal structure of SAHA complexed with HDAC2 (PDB ID: 4LXZ) shows the hydroxamate coordinated to the zinc ion [26]. The hydroxamate coordination is stabilized by hydrogen bonding to histidine-146 and tyrosine-306. Hydrophobic interactions occur between the phenyl ring of phenylalanine-155 and the aliphatic chain in SAHA. An additional hydrogen bond exists between aspartic acid-102 and the amide nitrogen of SAHA (Figure 3a). Jazz90 was docked 10 times into the ligand binding site of HDAC2 and of those, only 1 pose showed the hydroxamate coordinating to the zinc ion (Figure 3b). Interactions between Jazz90 and HDAC2 were similar to that seen for the crystal structure of the SAHA complex with hydrophobic interactions between the alkyl chain and phenylalanine-155 as well as the phenyl ring and proline-35. Four of the remaining poses resulted in an unexpected binding mode where the pyridinecarbothioamide group coordinated to the zinc ion (Figure 3c). This form of zinc coordination was verified via the Cambridge Structural Database (CCDC) with entry CCDC 1257652 showing coordination to zinc via a pyridinecarbothioamide. The phenyl ring of Jazz90 lays between phenylalanine-155 and phenylalanine-210 forming a π-stacking interaction (Figure 3c). The remaining poses were similar to the latter orientation; however, the pyridine nitrogen did not coordinate (ring rotated by 180°, not shown) and was therefore dismissed.

Acetylation of the histone-3 variant was assessed to understand the cellular effects of these compounds on HDAC inhibition. Jazz90 led to a significant 6.24- and 8.22-fold increase in deacetylation levels of histone-3 (H3) at concentrations of 1 and 4 μM, whereas Jazz167 caused a significant 6.3-fold increase in deacetylation levels of H3 at a concentration of 4 μM in PC3 cells (Figure 4b). For comparison, SAHA treatment resulted in a significant 12.9- and a 15.5-fold increase of acetylated H3 levels (Figure 4b). In contrast to PC3 cells, Jazz90 and Jazz167 failed to increase deacetylation in the DU145 cell line (Figure 4c). As HDAC overexpression is a common occurrence in prostate cancer, lower HDAC expression is seen in PNT1A cells [17]. Therefore, baseline acetylation levels in PNT1A cells are higher, as seen from the immunoblotting experiments. Interestingly, SAHA increased the deacetylation levels by a significant 2.1-fold and a 2.6-fold in PNT1A cells compared to the vehicle control (0.56 acetyl-H3/β-tubulin), whereas Jazz90 and Jazz167 did not induce significant acetylation (Figure 4d).

### 2.3. Compound-Mediated Cytostatic or Cytotoxic Mechanisms

Time-course analysis was conducted to determine if Jazz90 and Jazz167 were cytostatic. Jazz90 and Jazz167 led to 2.07- and 2.74-fold significant increases in the number of PC3 cells after 72 h at 4× EC_50_ values compared to time zero (Figure 5b,c). Similar results were obtained for SAHA, which led to a 1.82-fold increase in PC3 cell number at 4× EC_50_ value (Figure 5a). Interestingly, a similar increase in cell number was seen until 48 h for all treatment conditions (Figure 5). After this point, cell numbers reduced by 17, 10, and 30% in response to treatment with 4× EC_50_ concentrations of Jazz90, Jazz167, and SAHA, whereas the control cells increased by 49% from 48 to 60 h (Figure 5). In DU145 cells, SAHA, Jazz90 and Jazz167 led to 1.92-, 1.69-, and 2.02-fold increases in cell number at 1× EC_50_ values, respectively. In contrast, no significant difference in cell number was seen between compound treatments at 4× EC_50_ values over time in DU145 cells (Figure 5).

These results imply that Jazz90 and Jazz167 have a cytostatic effect in DU145 cells at 4× EC_50_ values, but not in PC3 cells. To understand the mechanism of action of these compounds further, cell cycle and apoptotic analyses were conducted. Jazz90 (6 μM) elicited a significant increase in G_0_/G_1_ cells by 23 and 14% at 48 and 60 h, respectively (Figure 6b,c), compared to SAHA, which increased the number of cells in the G_2_/M phase by 53 and 56% at 6 and 8 μM, respectively, at 24 h (Figure 6a). At 48 h, SAHA induced a significant G_2_/M phase arrest at 8 μM. At 60 h, no significant changes were observed in response to SAHA. This shows that the effects of Jazz90 are more persistent than SAHA. For DU145, significant differences in G_0_/G_1_ arrest were observed between all treatments and control at 48 h (Figure 7).

The cell cycle profile was further validated by determining changes in cell cycle checkpoint proteins, cyclins and cyclin-dependent kinase (CDK) inhibitors. At 24 h, a significant reduction of 35 and 53% in cyclin D levels compared to the control was seen in PC3 cells in response to Jazz90 at concentrations of 6 and 8 µM, respectively (Figure 8b). Similarly, SAHA showed a 35 and 69% decrease at the same concentrations; however, no significant decreases were observed for cyclin B (Figure 8c). Furthermore, CDK inhibitors were upregulated in response to compound treatment. Specifically, Jazz90 significantly increased p21 by 286% at a concentration of 8 µM (Figure 8d). However, Jazz167 did not change p21 levels (Figure 8d). Significant increases of 445 and 345% in p27kip1 were seen in response to Jazz90 and Jazz167 at a concentration of 8 µM. SAHA, in comparison, led to a significant increase of 366 and 480% in p27kip1 at concentrations of 6 and 8 µM (Figure 8e). A similar profile was obtained in DU145 cells, except that all compounds were more effective at inhibiting cyclin D and less effective at modulating p27kip1 in this cell line (Figure 9b).

Apoptosis induction was examined to determine if this was associated with the mechanism of action of the compounds. Minimal apoptosis (less than 8%) was seen in response to Jazz90, Jazz167, and SAHA in PC3 cells (Figure 9). No significant changes were observed in the expression of the antiapoptotic protein, bcl-2, between controls and compound treatments at 48 h in PC3 cells (Appendix A). Furthermore, necrosis was minimal for Jazz90 and Jazz167, but it peaked at 14 and 18% following treatment with 6 and 8 µM of SAHA (Figure 10). In DU145 cells, less than 2% of the cells were apoptotic at 24 and 48 h (Appendix A), and the percentage of necrotic cells did not exceed 11% for Jazz90, Jazz167, and SAHA (Appendix A). No significant differences were observed for bcl-2 between control and compound treatments at 48 h in DU145 cells (Appendix A).

### 2.4. Compound Withdrawal Effects

As the previous set of experiments showed a minimal level of apoptosis and necrosis, the persistent effects of the compounds were examined by withdrawing them and allowing the cells to grow for an additional 72 h. In PC3 cells, no significant differences in cell number over time was found for all treatments, while vehicle control cells significantly increased (Appendix A). However, DU145 cells significantly increased after 72 h of withdrawal of 1× EC_50_ values of SAHA, Jazz90 and Jazz167 (Appendix A).

## 3. Discussion

Jazz90 and its organorhodium derivative, Jazz167, have been identified as potent HDAC1, 6, and 8 inhibitors [22]. In this study, we investigated their anticancer potential in prostate cancer cell lines. Thioamides can react with nucleophiles and electrophiles and they also have an affinity for metal binding [27]. Given that HDACs feature a zinc (II) ion in the active site, ligand docking predicted that Jazz90 can bind to HDACs via either the hydroxamic acid or pyridinecarbothioamide moiety. Pyridinecarbothioamides have not been shown to have metalloenzyme inhibitory activity, and further assessment of the structure–activity relationship via replacing the hydroxamate acid in Jazz90 shall give a better insight into the potential of pyridine carbothioamide for zinc coordination in HDAC. Using the static model of ligand docking, similar hydrogen bond and hydrophobic interactions were seen in response to Jazz90 and SAHA. This was supported by their equipotent inhibition of HDAC activity in PC3 cells. Considering HDACs as potential targets for the pyridinecarbothioamide compounds, it is possible that Jazz90 and Jazz167 feature different binding modes to the Zn(II) ion. While Jazz167 resembles the HDAC inhibition of SAHA through targeting the Zn(II) ion with its hydroxamate moiety, Jazz90 may also bind through the pyridinecarbothioamide coordination motif (sulfur and nitrogen) to the Zn ion, as demonstrated by docking investigations. The Rh moiety to in Jazz167 impacts its physicochemical properties, including the lipophilicity and aqueous solubility, which may impact the anticancer potency.

Clinical data suggest that normal prostate epithelial cells have a lower HDAC activity, whereas PC3, LnCAP and DU145 cells have higher HDAC activity [17]. Jazz90 and Jazz167 are more cytotoxic against AR-null prostate cancer cells (PC3 and DU145) compared to normal prostate epithelial cells (PNT1A). They also have a higher selectivity index in PC3 cells in comparison to SAHA. These results are further supported by the significant inhibition of deacetylation of histone-3 in response to SAHA in PNT1A cells. In a previous study, SAHA inhibited deacetylation of histone-3 at concentrations of 2.5 and 5 μM [28]. Results from the selectivity index of Jazz90 and Jazz167 correlate with the levels of HDACs in LnCaP, PC3, and DU145 cells. This might explain the higher selectivity of Jazz90 and Jazz167 toward the AR-null (PC3 and DU145) cells in comparison to AR+ (LnCaP) cells. The cytotoxicity of SAHA in this study was comparable to previous studies. PC3 cells are relatively resistant to SAHA in comparison to DU145 and LnCaP cells [29]. The resistance can be attributed to the overexpression of bcl-2, an antiapoptotic protein in the PC3 cell line, whereas DU145 cells are reported to either lack or have a reduced expression of bcl-2 levels [29].

This study also aimed to define the cytostatic versus cytotoxic action of novel anticancer agents. Cell number, cell cycle phase changes, and apoptosis induction were used to determine this action. Previously, none of the studies associated with HDAC inhibitors have examined all of these parameters. Results from the time course analysis of the activity of SAHA, Jazz90, and Jazz167 in cells suggested that the compounds have a cytostatic action. Another reason for conducting a time-course analysis was to investigate an upregulation of resistance mechanisms. HDAC inhibitors have upregulated P-glycoproteins in colorectal cancer cell lines as early as 24 h after treatment [30,31]. However, no switch from negative to positive growth gradients occurred, which would have been observed if the compounds were being effluxed out of the cell. In fact, the opposite occurred in PC3 cells at 48 h. HDAC inhibitors can have different efficacies for different HDAC isoenzymes at different concentrations. A study conducted by Tang and colleagues showed that ACY-1215, despite being a HDAC6 inhibitor, display cross-selectivity to HDAC1 at higher concentrations [32]. Saturation of a HDAC isoenzyme can result in the binding of HDAC inhibitors to other isoenzymes. These results could be determined using posttranslational modification-based proteomic profiling using shotgun mass spectrometry.

Cell cycle arrest was observed in response to all three compounds, and the profiles were linked to a reduction in cyclin D1 and an increase in p21 and p27kip1. Minimal apoptosis induction (<3%) was observed in DU145 and PC3 cells, whereas necrosis elicited by Jazz90 and Jazz167 was less than 5%. In contrast, SAHA at 8 µM caused 13% of cells to undergo necrosis after 60 h, while 10% of DU145 cells were necrotic at 48 h. Similar results were observed in response to SAHA in previous studies. SAHA led to a G_2_/M phase arrest in PC3 cells and a G_0_/G_1_ phase arrest in DU145 cells [29,33]. Furthermore, Patra and colleagues (2013) showed an upregulation of p21 and p27 in response to SAHA in PC3 and DU145 cells [29]. Low rates of apoptosis of 14 and 17% were seen in DU145 and PC3 cells, respectively, in response to 9 μM of SAHA [34].

One of the reasons for the cell-specific effects may be that PC3 is PTEN- and p53-negative and DU145 have a loss-of-function mutated p53 [35,36,37]. Absence of PTEN and p53 is linked to proliferation and survival pathways in PC3 and DU145 cells. Another commonly mutated pathway in prostate cancer cells is the Erk protein, which following phosphorylation, triggers proliferation and survival. Effects of Jazz90 and Jazz167 are most likely mediated via reduction of the Erk phosphorylation, as seen in previous studies [6,38,39,40]. Overall, the results indicate that Jazz90 and Jazz167 can be classified as cytostatic. Cytostatic compounds have a role as they can lower the required doses of cytotoxic drugs when used in combination, which can reduce side effects [41]. Cytostatic drugs can also make cells more detectable by immune cells [42]. Furthermore, pyridinecarbothioamides have been shown to have metalloenzyme inhibitory activity, and further assessment of structure–activity relationships via replacing the hydroxamic acid moiety in Jazz90 will give a better insight into the potential of pyridinecarbothioamide–zinc coordination to induce biological effects. Thus, Jazz90 and Jazz167 should undergo further examination, including in animal models for toxicity and efficacy.

## 4. Materials and Methods

### 4.1. Materials

Prostate cancer cell lines, AR− (PC3 and DU145 cells) and AR+ (LnCAP cells) and mouse embryonic fibroblasts (NIH 3T3 cells) were obtained from American Type Culture Collection (Manassas, VA, USA). The epithelial prostate cell line immortalized with SV40 (PNT1A) was gifted from the Department of Anatomy, University of Otago. Primary antibodies to acetyl H3, cyclin D, cyclin B, p21, p27kip1, and bcl-2 were purchased from Cell Signaling Technology (Danvers, MA, USA). β-tubulin, β-actin, Dulbecco’s modified Eagle’s medium (DMEM) nutrient mixture Ham’s F-12, sulforhodamine B salt and propidium iodide (PI) were purchased from Sigma-Aldrich (Auckland, New Zealand). Acrylamide, bisacrylamide, sodium dodecylsulfate and PVDF membrane were purchased from Bio-Rad Laboratories (Hercules, CA, USA). Complete mini-EDTA-free protease inhibitor was purchased from Roche Diagnostics Corporation (Mannheim, Germany). Annexin V APC was obtained from BD Pharmingen (San Jose, CA, USA). FxCycle PI RNase staining solutions were ordered from Life Technologies (Christchurch, New Zealand). The Histone Deacetylase (HDAC) Activity Assay Kit (Fluorometric) was purchased from Abcam (Melbourne, Australia). Jazz90 and Jazz167 were synthesized as previously described [23]. SAHA was obtained from AK Scientific (Union City, CA, USA).

### 4.2. Cell Maintenance

PC3 and DU145 cells were maintained in 5% DMEM/Ham’s F12 supplemented with 100 units/mL penicillin, 100 units/mL of streptomycin and 2.2 g/L of NaHCO_3_, whereas LnCAP and PNT1A cells were cultured in RPMI640 media with 10% FBS, 100 units/mL penicillin. All cells were maintained at 37 °C in a humidified atmosphere of 5% CO_2_.

### 4.3. Cytotoxicity Assays and Time-Course Analysis

PC3 (4 × 10^3^ cells/well), DU145 (5 × 10^3^ cells/well), LnCAP (1 × 10^4^ cells/well), PNT1A (1 × 10^4^ cells/well), and NIH 3T3 cells (3.5 × 10^3^ cells/well) were plated in 96-well plates. After 24 h, the cells were treated with Jazz90, Jazz167, and SAHA at concentrations ranging from 0 to 150 μM. Vehicle control cells were treated with 0.5% DMSO. The cells were fixed using 10% trichloroacetic acid (TCA) after 72 h. The sulforhodamine B (SRB) assay was then used to determine the cell number as previously described [43]. EC_50_ values were determined by nonlinear regression using Prism9 software. Three independent experiments were carried out in triplicate.

### 4.4. HDAC Inhibition

The Histone Deacetylase Activity Assay Kit (Fluorometric; ab156064) was used to measure HDAC inhibition. Corning black, 96-well plates were used and the following groups were run: no enzyme control, solvent control, and 0.05, 0.1, and 0.2 μM of Jazz90, Jazz167, and SAHA. Trichostatin A (TSA) was used as a positive control as provided with the kit. The principle of the assay relies on the deacetylation of the fluoro-substrate peptide by HDACs. Fluoro-deacetylated peptides are produced in the process, which can then be hydrolyzed by lysyl endopeptidases into lysine and 7-amino-4-methylcoumarin (AMC). AMC leads to an increase in fluorescence intensity. The experiment was run for 70 min and readings were taken every 2 min using a SpectraMax i3X fluorimeter (San Jose, CA, USA). The fluorescence intensity was read at an excitation/emission wavelength of 360 nm/450 nm. Three independent experiments were carried out using HeLa and PC3 nuclear lysates.

### 4.5. HDAC Docking

Jazz90 was drawn using Avogadro, after which the energy of the geometry was optimized. The structure of HDAC2 complexed with SAHA (PDB ID: 4LXZ) was obtained from the Research Collaboratory for Structural Bioinformatics (RCSB) Protein Data Bank [26]. Molecular docking was carried out using GOLD software (Cambridge Crystallographic Data Centre, Cambridge, UK). The binding site was defined as within 8 Å and a metal distance constraint was set to a distance of 3.5 Å. Jazz90 was then docked onto the crystal structure and was assessed for its binding orientation.

### 4.6. Preparation of Sample Lysates for Western Blotting

PC3 and DU145 cells were seeded in 100 × 20 mm Petri dishes at 1 × 10^6^ and 1.5 × 10^6^ cells per dish, respectively, with 10 mL of DMEM/HamF12 supplemented with 5% FBS and 100 U/mL penicillin, 100 μg/mL streptomycin and 2.2 g/L NaHCO_3_. PNT1A cells were seeded at 2 × 10^6^ cells with 10 mL of RPMI640 media with 10% FBS and 100 units/mL penicillin. Cells were treated after 24 h with 1, 4, 6, and 8 μM of SAHA, Jazz90 and Jazz167. At the end of the treatment, whole cell lysates were obtained. The protein concentration of these lysates was determined using the bicinchoninic acid method [44].

### 4.7. Western Blotting

Samples containing 10 μg of protein from cellular, nuclear and cytoplasmic extractions were resolved using SDS-PAGE at 100 V. After the sample ran down to the bottom of the gel, the membrane was removed and transferred into the transfer buffer. A sandwich containing equilibrated fiber pad and blotting paper and an activated membrane was made in cassettes. To stop the apparatus from overheating, ice was placed in the apparatus. A voltage of 100 V was set, and the transfer process was carried out for 90 min. The membrane was blocked with BSA blocking buffer 1×, followed by a primary antibody incubation (acetyl-H3, cyclin D, cyclin B, p21, p27kip1, bcl-2, β-tubulin, and β-actin) overnight, after which the membrane was washed six times with TBST and incubated with secondary antibody for a period of one hour. After six further washes with TBS, chemiluminescent solutions were then added to the membrane, and X-ray films were exposed to the membrane, after which the films were developed. The films were analyzed using a BioRad GS710 densitometer (Hercules, CA, USA), and the protein density was calculated as a percentage of β-actin or β-tubulin. Three independent experiments were carried out.

### 4.8. Cell Cycle Analysis

Cell cycle distribution was assessed using propidium iodide (PI), which measures the DNA content in the cells. PC3 (1 × 10^6^ cells) and DU145 (1.5 × 10^6^) cells were seeded in 100 × 20 mm Petri dishes. After allowing the cells to attach to the surface for a period of 24 h, the cells were treated with either 6 or 8 μM of the compounds (SAHA, Jazz90 and Jazz167) or vehicle control (0.5% DMSO). Based on the results from the time-course experiments, time-points were chosen for sample extraction. For PC3 cells, the cell cycle distribution was analyzed at 24, 48, and 60 h, whereas for DU145 cells, it was performed at 24 and 48 h. Supernatants were collected and cells were harvested and washed with PBS. Cells were then fixed in 70% ethanol. Subsequently, cells were rehydrated with PBS and stained with PI RNase in the dark at 4 °C. The samples were analyzed using the BC Gallios flow cytometer (Beckman Coulter, Auckland, New Zealand). The FlowJo software was used to determine the percentage of cells in each of the phases of the cell cycle. Results were expressed as a percentage of cell number in each of the cell cycle phases. Three independent experiments were carried out.

### 4.9. Apoptosis Analysis

PC3 (1 × 10^6^ cells) and DU145 (1.5 × 10^6^ cells) were seeded in 100 × 20 mm Petri dishes. After 24 h, the cells were treated with 6 and 8 μM of SAHA, Jazz90, and Jazz167 or vehicle control (0.5% DMSO). Apoptosis was analyzed at the same time points as the cell cycle distribution. A similar procedure to harvest the cells as in the cell cycle analysis experiments was used. The cells were then suspended in Annexin-V APC for a period of 5 min, followed by PI staining. The samples were analyzed using the BC Gallios flow cytometer. The data were analyzed using Kaluza software (Beckman Coulter, Auckland, New Zealand). Three independent experiments were carried out.

### 4.10. Statistical Analysis

Time course experiments were assessed using a two-way ANOVA followed by Bonferroni’s post hoc test. For experiments that did not have time as an independent variable, one-way ANOVA was used, followed by Bonferroni’s post hoc test. *p* < 0.05 was the minimum requirement for a significant difference.

## Figures and Tables

**Figure 1 pharmaceuticals-14-00103-f001:**
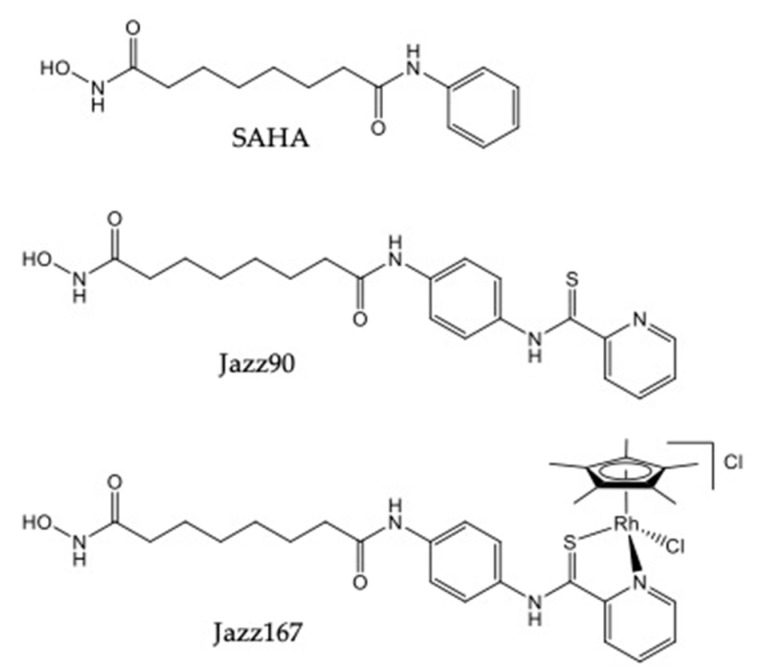
Structures of SAHA, Jazz90, and Jazz167.

**Figure 2 pharmaceuticals-14-00103-f002:**
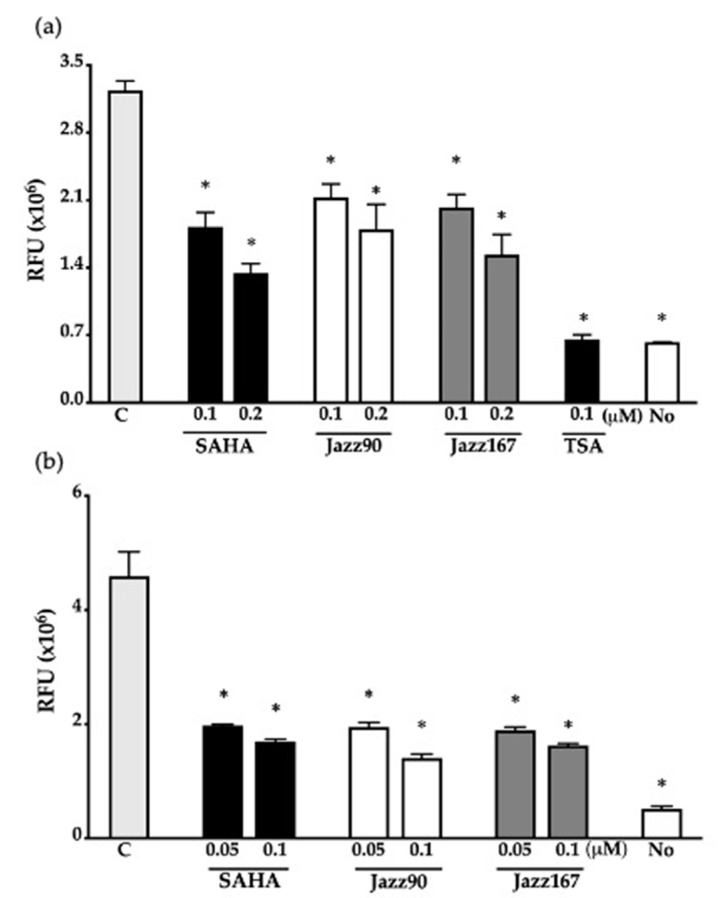
HDAC inhibition elicited by Jazz90 and Jazz167. (**a**) HeLa nuclear lysates were treated with 0.1 and 0.2 µM of Jazz90, Jazz167, and SAHA. TSA (0.1 µM) was used as a positive control and a no enzyme control (No) was also used. Vehicle control (C) lysates were treated with 0.5% DMSO. (**b**) PC3 nuclear lysates were treated with 0.05 and 0.1 µM of SAHA, Jazz90, and Jazz167. Bar graphs represent the relative fluorescence units (RFU) measured after 70 min of treatment. A two-way ANOVA was conducted followed by Bonferroni’s post hoc test. * indicates significant difference to control, *p* < 0.05. Images from the kinetic assays are provided in Appendix A.

**Figure 3 pharmaceuticals-14-00103-f003:**
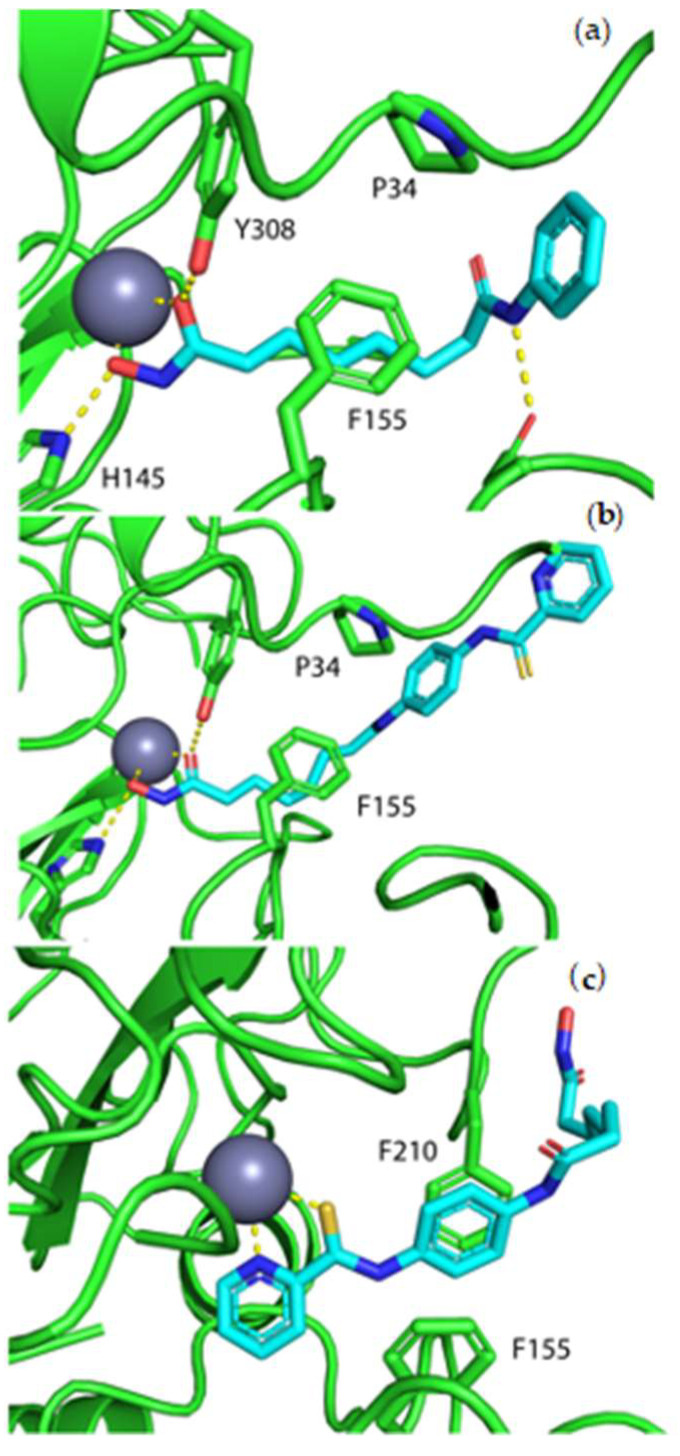
SAHA and Jazz90 binding to HDAC2. Jazz90 was docked into HDAC2 (PDB ID: 4LXZ). (**a**) Crystal structure of SAHA (cyan) complexed with HDAC2, (**b**) Jazz90 docked into HDAC2 with hydroxamate coordination to zinc, and (**c**) Jazz90 docked into HDAC2 with pyridinecarbothioamide coordination. Green ribbons and sticks represent the active site of HDAC2. (Blue atoms = nitrogen, yellow = sulfur, and red = oxygen).

**Figure 4 pharmaceuticals-14-00103-f004:**
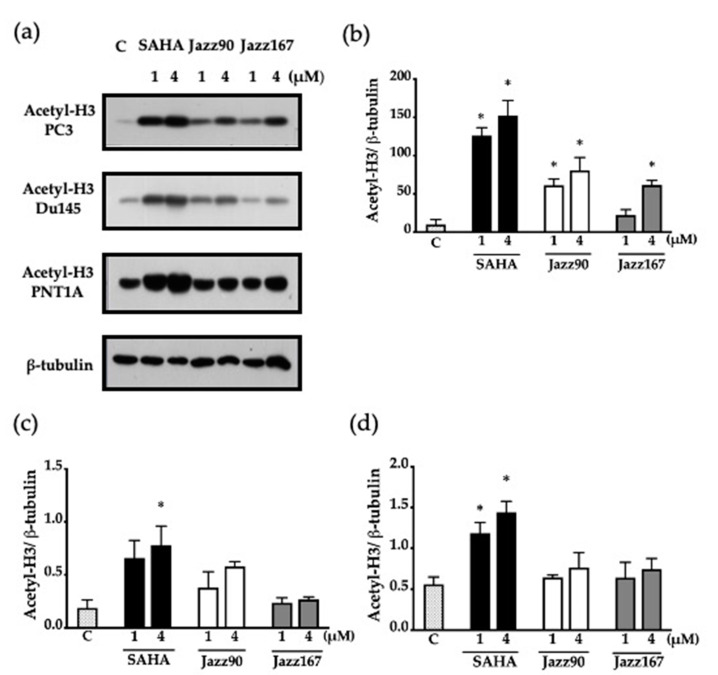
Acetylation levels of CRPC (PC3 and DU145) and normal prostate (PNT1A) cells after treatment at concentrations of 1 and 4 µM of SAHA, Jazz90 and Jazz167. Vehicle control cells (C) were treated with 0.5% DMSO. Cells were harvested 24 h after treatment. (**a**) Representative Western blots for acetyl-H3 and β-tubulin (loading control) are shown for each of the cell lines. Scanning densitometry of Western blots for (**b**) PC3, (**c**) DU145, and (**d**) PNT1A cells. Bars represent the mean ± S.E.M. from three independent experiments. Data were analyzed using one-way ANOVA followed by Bonferroni’s post hoc test; * indicates significant increases relative to the control, *p* < 0.05.

**Figure 5 pharmaceuticals-14-00103-f005:**
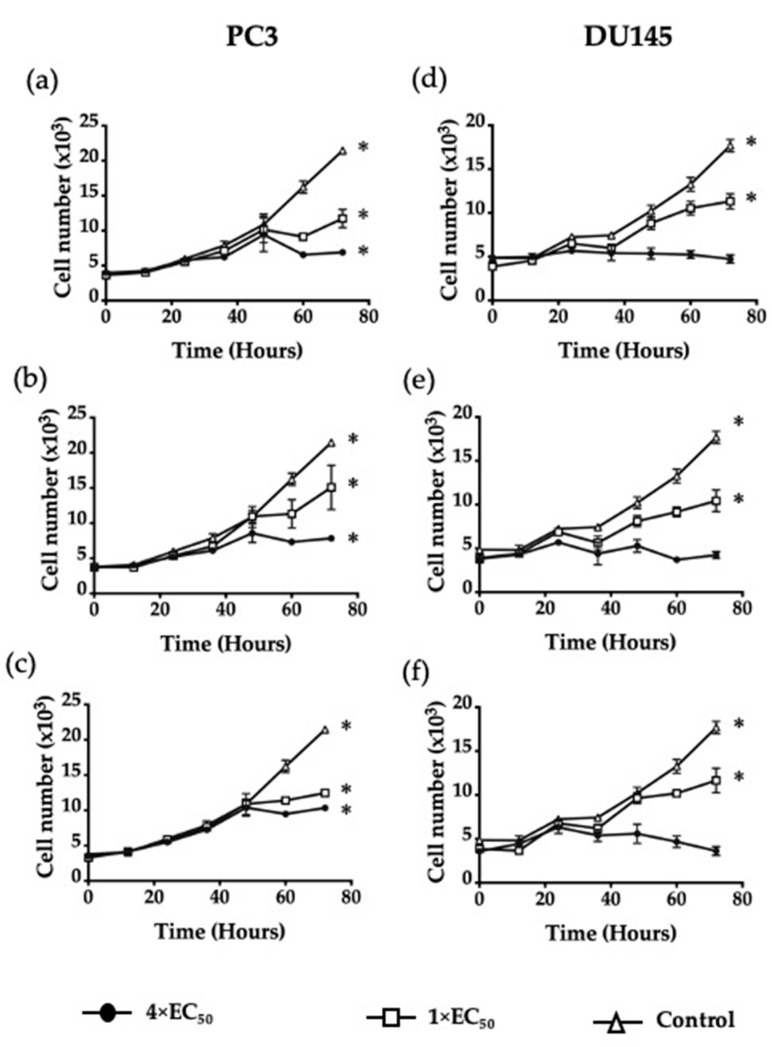
Time-course cytotoxicity analysis in prostate cancer cells. PC3 cells were treated with (**a**) SAHA, (**b**) Jazz90, and (**c**) Jazz167, and DU145 cells were treated with (**d**) SAHA, (**e**) Jazz90, and (**f**) Jazz167 at concentrations of 1× EC_50_ and 4× EC_50_ for 0–72 h. Control cells were treated with 0.5% DMSO. Cell number was measured using the SRB assay. Symbols indicate cell number ± S.E.M from three independent experiments performed in triplicate. Data were analyzed using a two-way ANOVA coupled with Bonferroni’s post hoc test; * indicates significant differences compared to the start of the experiment, *p* < 0.05.

**Figure 6 pharmaceuticals-14-00103-f006:**
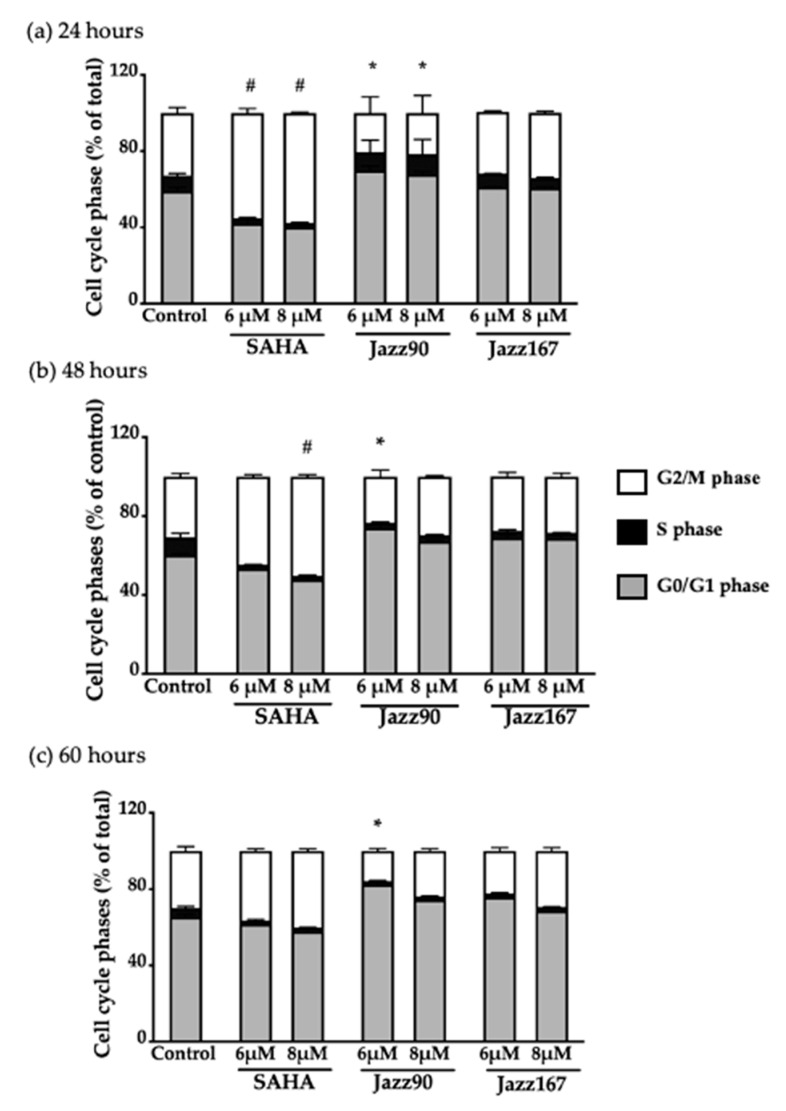
Cell cycle phases of PC3 cells at different time-points. PC3 cells were treated with SAHA, Jazz90 and Jazz167 at concentrations of 6 and 8 µM for (**a**) 24, (**b**) 48, and (**c**) 60 h. Vehicle control cells were treated with 0.5% DMSO. Cell cycle phases were measured using flow cytometry. Bars represent the mean ± S.E.M. of the percentage of cells in each of the cell cycle phases (G_0_/G_1_, S and G_2_/M phases) from three independent experiments. Data were analyzed using a two-way ANOVA coupled with Bonferroni’s post hoc test. # indicates G_2_/M significantly different from the control, *p* < 0.05; * indicates G_0_/G_1_ significantly different from the control, *p* < 0.05.

**Figure 7 pharmaceuticals-14-00103-f007:**
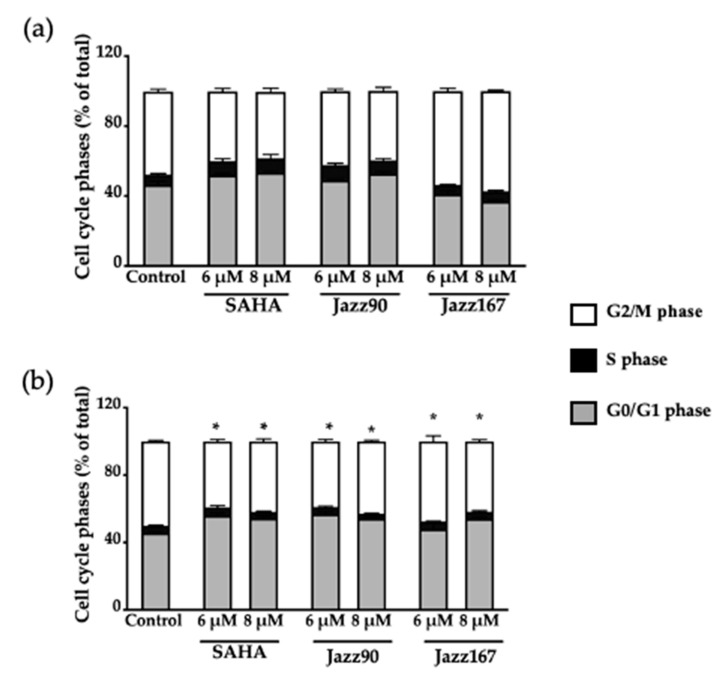
Cell cycle phases of DU145 cells at different time-points. DU145 cells were treated with SAHA, Jazz90 and Jazz167 at concentrations of 6 and 8 µM for (**a**) 24 and (**b**) 48 h. Vehicle control cells were treated with 0.5% DMSO. Cell cycle phases were measured using flow cytometry. Bars represent the mean ± S.E.M. of percentage of cells in each of the cell cycle phases (G_0_/G_1_, S, and G_2_/M phases) from three independent experiments. Data were analyzed using a two-way ANOVA coupled with Bonferroni’s post hoc test. * indicates G_0_/G_1_ significantly different from the control, *p* < 0.05.

**Figure 8 pharmaceuticals-14-00103-f008:**
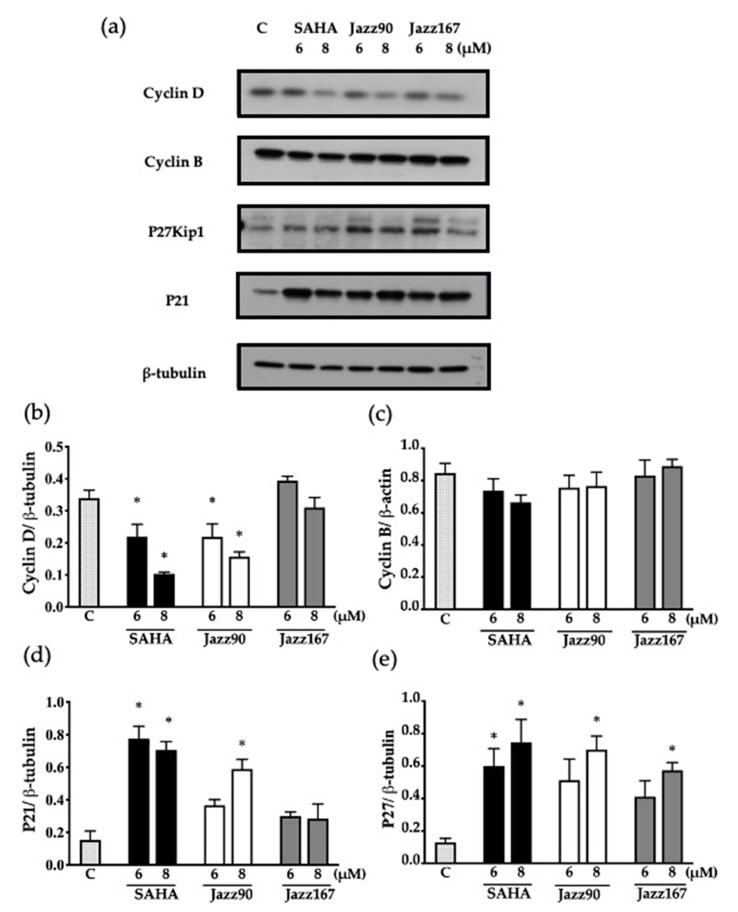
Cell cycle protein expression in PC3 cells. Cyclin D, cyclin B, p27kip1 and p21 levels were assessed in response to SAHA, Jazz90, and Jazz167 in PC3 cells at concentrations of 6 and 8 µM for 24 h. Vehicle control cells (C) were treated with 0.5% DMSO. Cell lysates were analyzed using Western blotting and β-tubulin was used as the loading control. (**a**) Representative blots for each of the proteins. Scanning densitometry graphs for (**b**) cyclin D, (**c**) cyclin B, (**d**) p21, and (**e**) p27kip1 are depicted. Bars represent the means ± S.E.M. from three independent experiments. Data were analyzed using a one-way ANOVA coupled with Bonferroni’s post hoc test; * indicates significant differences relative to the control, *p* < 0.05.

**Figure 9 pharmaceuticals-14-00103-f009:**
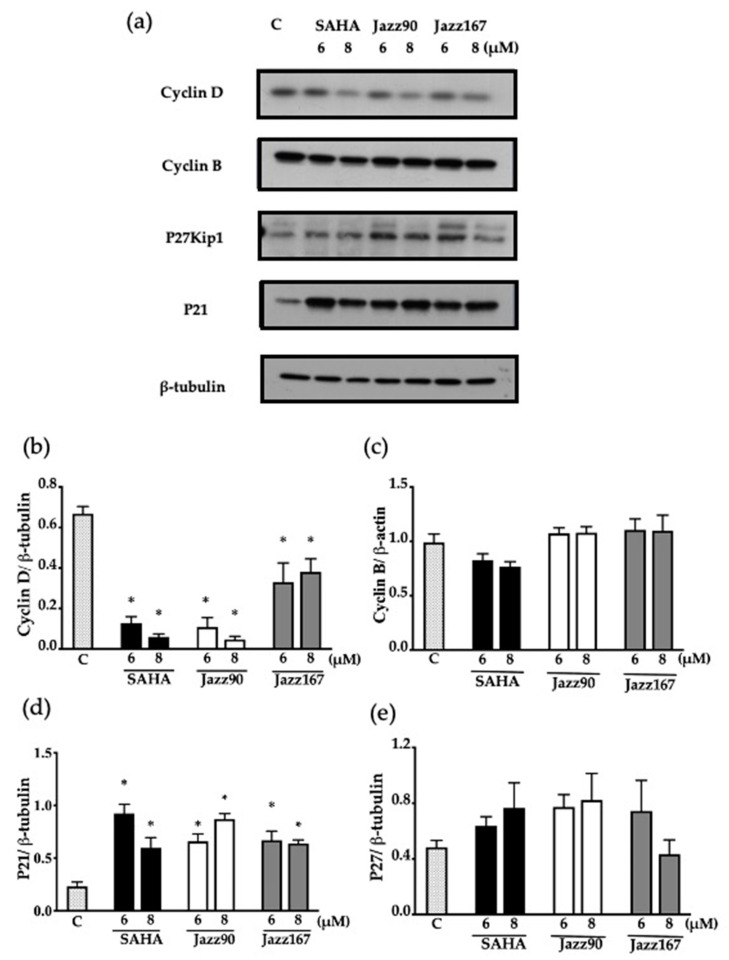
Cell cycle protein expression in Du145 cells. Cyclin D, cyclin B, p27kip1 and p21 levels were assessed in response to SAHA, Jazz90 and Jazz167 in PC3 cells at concentrations of 6 and 8 µM for 24 h. Vehicle control cells (C) were treated with 0.5% DMSO. Cell lysates were analyzed using Western blotting and β-tubulin was used as the loading control. (**a**) Representative blots for each of the proteins. Scanning densitometry graphs for (**b**) cyclin D, (**c**) cyclin B, (**d**) p21, and (**e**) p27kip1 are depicted. Bars represent the means ± S.E.M. from three independent experiments. Data were analyzed using a one-way ANOVA coupled with Bonferroni’s post hoc test; * indicates significant differences relative to the control, *p* < 0.05.

**Figure 10 pharmaceuticals-14-00103-f010:**
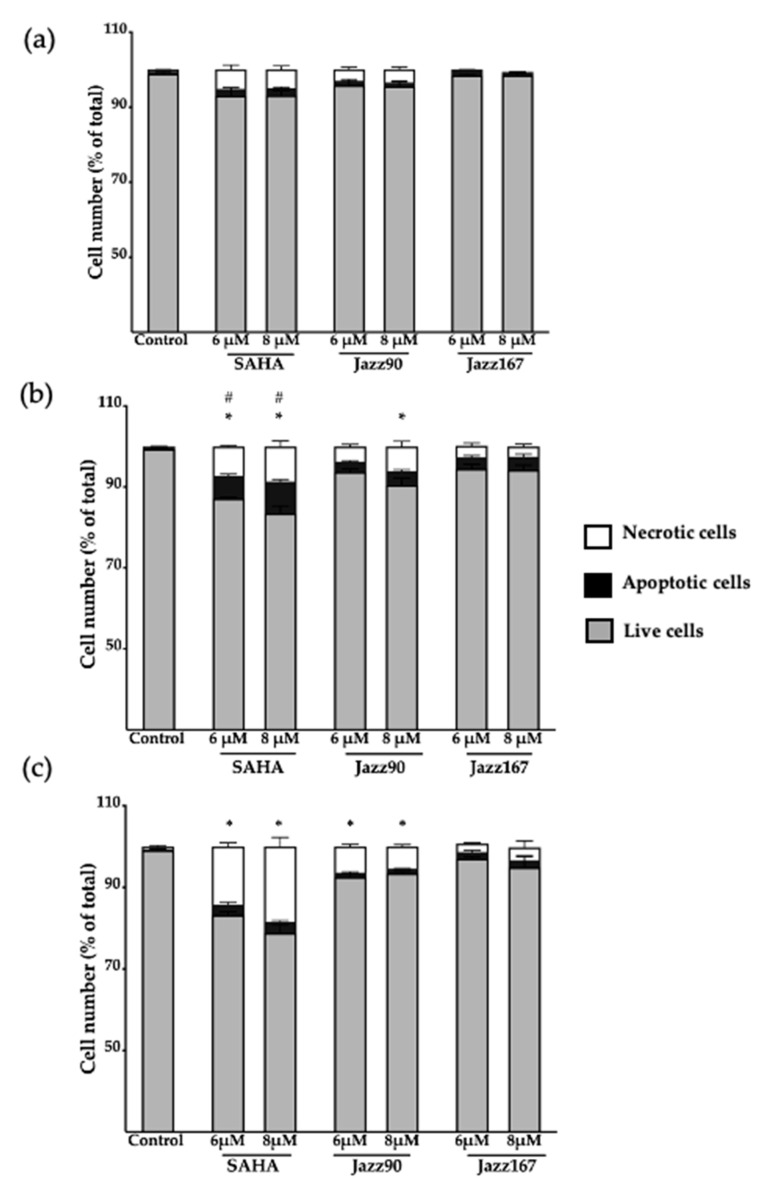
Analysis of apoptosis and necrosis in PC3 cells. Cells were treated with SAHA, Jazz90 and Jazz167 at concentrations of 6 and 8 µM. Vehicle control cells were treated with 0.5% DMSO. Cells were harvested at (**a**) 24, (**b**) 48, and (**c**) 60 h, after which the cells were stained with Annexin V and PI, and analyzed by flow cytometry. Bars represent the mean proportion of live, apoptotic and necrotic cells ± S.E.M. from three independent experiments. Data were analyzed using a two-way ANOVA coupled with Bonferroni’s post-hoc test; * indicates significant differences in necrotic cells relative to the control, *p* < 0.05; # indicates significant differences in apoptotic cells relative to the control, *p* < 0.05.

**Table 1 pharmaceuticals-14-00103-t001:** Selectivity index of Jazz90 and Jazz167.

Compound	Cancer/Preneoplastic Cell Lines	EC_50_ Values (μM) (Cancer Cells)	EC_50_ Values (μM) (PNT1A Cells)	Selectivity Index ^1^
Jazz90	PC3	2.09 ± 0.33	6.13 ± 0.74	2.93
	DU145	2.50 ± 0.65		2.45
	LnCaP	3.04 ± 0.93		2.02
	NIH 3T3	3.93 ± 0.65		0.64
Jazz167	PC3	2.39 ± 0.16 *	8.36 ± 0.10	3.50
	DU145	6.44 ± 1.26 *		1.30
	LnCaP	3.36 ± 0.71		2.53
	NIH 3T3	16.79 ± 1.77 *		0.50
SAHA	PC3	1.13 ± 0.11	1.87 ± 0.21	1.65
	DU145	0.69 ± 0.08		2.71
	LnCaP	0.61 ± 0.08		3.07
	NIH 3T3	1.41 ± 0.21		1.33

^1^ Selectivity index was calculated by dividing the EC_50_ values in normal PNT1A cells by that in cancer cells. * Significantly different from SAHA, *p* < 0.05. Data were analyzed using a one-way ANOVA followed by Bonferroni’s post-hoc test.

## Data Availability

The data presented in this study are available within the article, associated Appendix A, or on request from the corresponding author.

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
