# Peer review of "Cytostatic Action of Novel Histone Deacetylase Inhibitors in Androgen Receptor-Null Prostate Cancer Cells"

_pharmaceuticals, 2021, doi:10.3390/ph14020103_

Round 1

Reviewer 1 Report

The manuscript titled ‘Cytostatic action of novel histone deacetylase inhibitors in androgen receptor-null prostate cancers’ studied the effects of two HDAC inhibitors (Jazz90 and Jazz167) on castration-resistant prostate cancer. As such, the experiments are appropriately conducted and well written. However, there are few comments that need to be addressed: 

  1. Although AR-null aspect is heavily emphasized in the title and introduction, authors didn’t follow it through in the discussion. This needs attention to connect the relevance of the results with the primary goal of the manuscript (i.e., how HDAC inhibitors can work in AR-null prostate cancer)
  2. SAHA (Vorinostat) has been used as a comparator but, except ref. 31, previous work was of this molecule was not discussed either in introduction or discussion.
  3. 2 and some of the other ones, the SEM is visible in some but not all the data points. Please confirm this is correct.
  4. There are too many figures in the manuscript. Authors should consider consolidating some of them or include as supplementary material
  5. 31 is clubbed together with Ref. 30.
  6. Title: instead of “..prostate cancers”, it should say “..prostate cancer cells”.
  7. Table 1: is there any statistical significance between the treatments?
  8. Justification is needed about the selection of concentrations of the compounds in the study. Are they physiologically achievable?

Author Response

  1. Although AR-null aspect is heavily emphasized in the title and introduction, authors didn’t follow it through in the discussion. This needs attention to connect the relevance of the results with the primary goal of the manuscript (i.e., how HDAC inhibitors can work in AR-null prostate cancer)

This has now been addressed in the introduction and discussion section. The introduction now contains…

Chaio and colleagues (2009) also showed that AR-null cell lines, PC3 and DU145, possess 3 - 4x higher HDAC activity in comparison to the AR+ (LnCaP) cell line [17]. Therefore, HDAC inhibitors are likely to be more useful in an AR-null environment in comparison to an AR+ environment.

The discussion section now contains…

Clinical data suggests that normal prostate epithelial cells have a lower HDAC activity, whereas PC3, LnCAP and DU145 cells have higher HDAC activity [17]. Jazz90 and Jazz167 were more cytotoxic against AR-null prostate cancer cells (PC3 and DU145) as compared to normal prostate epithelial cells (PNT1A). They also had a higher selectivity index in PC3 cells in comparison to SAHA. These results are further supported by the significant inhibition of deacetylation of histone-3 in response to SAHA in PNT1A cells. In a previous study, SAHA inhibited deacetylation of histone-3 at concentrations of 2.5 and 5 M [28] Results from the selectivity index of Jazz90 and Jazz167 correlate with the levels of HDACs in LnCaP, PC3 and DU145 cells. This might explain the higher selectivity of Jazz90 and Jazz167 towards the AR-null (PC3 and DU145) cells in comparison to AR+ (LnCaP) cells. 

  1. SAHA (Vorinostat) has been used as a comparator but, except ref. 31, previous work was of this molecule was not discussed either in introduction or discussion.

Previous studies including the effects of SAHA on cell cycle, cell cycle proteins, apoptosis and acetylation of histone-3 have been cited and referenced in both the introduction and discussion sections.

The introduction now contains… The poor efficacy of HDAC inhibitors in clinical trials for CRPC are likely to be attributed to the lack of assessment of AR expression. HDAC inhibitors (romidepsin, SAHA, panobinostat and pracinostat) are well-tolerated by patients and have been FDA approved for hematological malignancies and a HDAC inhibitor (chidamide) has cleared phase III clinical trials in combination with exemestane for locally advanced or metastatically recurrent hormone receptor-positive-breast cancer.

The discussion section now contains… In a previous study, SAHA inhibited deacetylation of histone-3 at concentrations of 2.5 and 5 M [28] Results from the selectivity index of Jazz90 and Jazz167 correlate with the levels of HDACs in LnCaP, PC3 and DU145 cells. This might explain the higher selectivity of Jazz90 and Jazz167 towards the AR-null (PC3 and DU145) cells in comparison to AR+ (LnCaP) cells. 

Similar results were observed in response to SAHA in previous studies. SAHA led to a G2/M phase arrest in PC3 cells and a G0/G1 phase arrest in DU145 cells [29,33]. Furthermore, Patra and colleagues (2013) showed an upregulation of p21 and p27 in response to SAHA in PC3 and DU145 cells [29]. Low rates of apoptosis of 14% and 17% were seen in DU145 and PC3 cells, respectively, in response to 9 M of SAHA [34]. 

  1. In Figure 2 and some of the other ones, the SEM is visible in some but not all the data points. Please confirm this is correct.

Yes, that is correct. This is because the SEM is contained within the symbol (i.e., it is not larger than the range encompassed by the symbol).

  1. There are too many figures in the manuscript. Authors should consider consolidating some of them or include as supplementary material.

Figure 2 has been moved to supplementary Figure 1 and Figure 12 has also been moved to supplementary Figure 9.

  1. 31 is clubbed together with Ref. 30.

References have been updated.

  1. Title: instead of “..prostate cancers”, it should say “..prostate cancer cells”.

The title has been changed.

  1. Table 1: is there any statistical significance between the treatments?

The significant differences are now indicated in Table 1.

  1. Justification is needed about the selection of concentrations of the compounds in the study. Are they physiologically achievable?

SAHA was orally administered at 400 mg in the clinical trial that led to its approval in peripheral cutaneous T-cell lymphoma. Pharmacokinetic studies from phase I trials showed that the highest plasma concentration of approximately 667 ng/ml was observed at 400 mg of SAHA when administered orally. However, intravenous administration of SAHA led to 2.31 mg/ml. On converting 6 and 8 mM of SAHA to physiological concentrations, values of 1.58 mg/ml and 2.11 mg/ml, respectively, were calculated, which shows that the highest concentration exceeds by a factor of approximately 2 or 3 if administered orally and falls within the limit if administered intravenously (Kelly et al., 2005). The next phase of our study will assess the pharmacokinetic and toxicity parameters for Jazz90 and Jazz167 in mice following oral gavage, which will give us an indication if these concentrations can be achieved physiologically.

Kelly et al. Phase I Study of an Oral Histone Deacetylase Inhibitor, Suberoylanilide Hydroxamic Acid, in Patients With Advanced Cancer. (2005)

Reviewer 2 Report

This work by Rana Z. et al. is just an extention of their recent published article (i.e. Ref.#22; Angew Chem Int Ed. 2020), wherein it has been shown that the two compounds at issue, namely Jazz90 and Jazz167, inhibited at nanomolar level the isoform 6 of HDACs and exerted antiproliferative activity in the range low-micromolar to sub-micromolar against a panel of tumor cell lines. The authors basically extended their studies by testing these two compounds, structurally related to SAHA, against both AR+ (LnCaP) and AR- (PC3 and DU145) prostate cancer cell lines (PC3 and DU145). They also investigated the cytostatic potential versus the cytotoxic effect of these two compounds in prostate cancer cells, in order to get insights about their mechanism of action and clinical potential. 

Overall the work is well described but I have to say that somehow mismatches with their previous studies, at least in some points.

1) As HDAC-1, HDAC-2 and HDAC-3 are the isoforms mostly implicated in prostate cancer, they should have focused on the selectivity towards these isoforms instead of using nuclear lysates to check a pan-HDAC inhibition, possibly providing di IC50 values and molecular docking analysis.

2) The discussion paragraph is quite long and in some points speculative. For instance, the reference to vitamins B as well as the drug pinacidil (ref. 28-30) is useless. This section must be shortened and focused on the obtained results and conclusions/perspectives. I suggest to move part of it to the Introduction section, especially when you talk about the description of the coumpounds.

3) Ref.#31 must be numbered within the reference list. 

Author Response

  1. As HDAC-1, HDAC-2 and HDAC-3 are the isoforms mostly implicated in prostate cancer, they should have focused on the selectivity towards these isoforms instead of using nuclear lysates to check a pan-HDAC inhibition, possibly providing di IC50 values and molecular docking analysis.

HDAC-1, -2, -3 and -8 are nuclear HDACs. Wang and colleagues (2008) showed that HDAC-1-5 were upregulated in DU145 and PC3 cells. Therefore, nuclear extraction assessed the activity of drugs on the three isoforms, HDAC-1,-2 and -3. Also, in this study, the nuclear lysate for PC3 cells was used. In the previous study by Hanif et al (2020), HDAC1, HDAC6 and HDAC8 human recombinant enzymes were used, which does not reflect a cellular environment as HDACs function in complexes, and different complexes can result in HDACs that have different functions under different cellular environments. Furthermore, complex formation has been shown to interfere with inhibitor affinities (Decluve et al., 2012).

Decluve et al. Roles of histone deacetylases in epigenetic regulation: emerging paradigms from studies with inhibitors (2012).

  1. The discussion paragraph is quite long and in some points speculative. For instance, the reference to vitamins B as well as the drug pinacidil (ref. 28-30) is useless. This section must be shortened and focused on the obtained results and conclusions/perspectives. I suggest to move part of it to the Introduction section, especially when you talk about the description of the compounds.

The points have been addressed. Information about the compounds have been moved to the Introduction section. The references to vitamin B and pinacidil were made to highlight that pyridine groups as a moiety in itself is not toxic as it has been incorporated in various other compounds. Overall, the discussion section has been modified taking in the points raised by both reviewers.

  1. #31 must be numbered within the reference list. 

This has been fixed.

Round 2

Reviewer 2 Report

The article has been sufficiently revised and the authors provided plausible explanations about  the issues previously raised and added new convincing references.